# Tensile, Compressive, and Flexural Characterization of CFRP Laminates Related to Water Absorption

**Sudarisman Sudarisman [1,\*], Haniel Haniel [1], Angger Kaloka Taufik [1], Muhammad Tiopan [1], Rela Adi Himarosa [1] and Muhammad Akhsin Muflikhun [2,3]**

1 Department of Mechanical Engineering, Universitas Muhammadiyah Yogyakarta, Jl. Brawijaya, Yogyakarta 55183, Indonesia; haniel203@gmail.com (H.H.); akaloka80@gmail.com (A.K.T.); tiopan37@gmail.com (M.T.); rela.himarosa@umy.ac.id (R.A.H.)
2 Department Mechanical and Industrial Engineering, Gadjah Mada University, Komplek UGM Bulaksumur, Yogyakarta 55281, Indonesia; akhsin.muflikhun@ugm.ac.id
3 Center for Advanced Manufacturing and Structural Engineering (CAMSE), Gadjah Mada University, Komplek UGM Bulaksumur, Yogyakarta 55281, Indonesia
\* Correspondence: sudarisman@umy.ac.id

**Abstract:** CFRP structures are often exposed to humid environment resulting in water absorption and causing property degradation. Water swelling and its effect on tensile, compressive, and flexural properties were investigated according to ASTM standards. Fracture modes were evaluated by analyzing micrographs of fracture areas. The specimens were cut from twill wave CFRP composite plates fabricated using a vacuum infusion technique. Some of them were immersed in water prior to being mechanically tested. It was found that tensile strength, as well as compressive, and flexural strength and moduli decreased due to water swelling, but fracture strain was found to increase due to water swelling. The most severely affected by water swelling is flexural strength (decreased by 25.72%), and the least is compressive modulus (decreased by 1.89%). Tensile specimens underwent fibre breakage followed by matrix cracking, compressive and flexural specimens showed fibre buckling followed by kinking and crushing where flexural specimens failed in their compressive side. In conclusion, water absorption has a bad impact on the composite strength.

**Keywords:** CFRP; water swelling; tension; compression; flexure; fibre kinking; crushing

## 1. Introduction

The use of composite materials in daily life has gained great acceptance, varying from producing household appliances, engine components, ships, and car bodies made of polymer composite materials. The advantages of composite materials over metal-based materials can be found in their superior properties such as high corrosion resistance, outstanding strength-to-weight ratio, and excellent results in static and fatigue tests [1,2]. Other properties of composites include lower density and higher hardness [3,4]. If more parts of a vehicle are made of composites, the overall weight of the vehicle will be lower which results in a better fuel efficiency [5]. The advantages of this composite material are suitable for use in the automotive, shipping and aircraft industries [6,7].

Carbon fibre is known for its high tensile strength, low density, low thermal and electrical conductivity and good creep resistance [8,9]. Its structure and properties are considerably stable under extreme fluid pressure, and temperature conditions. The addition of carbon fibre can also increase the attenuation of sound and vibration in the cabin to provide comfort and safety to the passengers due to carbon fibre low density, tensile modulus, and its resistant to high temperatures [10]. However, carbon fibre has a low compressive strength [11,12]. To overcome this problem, carbon fibre is combined with polymer to produce strong carbon fibre reinforced polymers (CFRP) composites. When a thermoset polymer is used, the manufacturing process can be carried out at room temperature by

considering the chemical substances used as controllers of cross-link polymerization to obtain optimum results. Epoxy, a kind of thermosetting, polymer belongs to a group of polymers that are used as coatings and matrices in producing composite materials in several structures. This resin is also used as a mixture of packaging materials, moulding materials, and adhesives [13,14].

Epoxy structure is amorphous, and its atoms are strongly bonded, it cannot be melted, and cannot be recycled. The advantages of epoxy are resistance to heat, moisture and chemicals, and good mechanical, insulating and adhesive properties. In addition, it is also easy to be modified and manufactured. However, epoxy also has disadvantages, e.g., being sensitive to water and brittle. The use of epoxy as matrix material is limited by its low toughness and brittleness [14,15]. Several studies on the use of composites as vehicle components have been carried out. Sunardi, et al. [16] reported that the tensile strength of pandanus leaf fibre composites for motor vehicle body applications showed the highest tensile strength (20.741 N/mm$^2$) was obtained from that with vertical fibre direction, while the lowest (17.955 N/mm$^2$) was obtained from that with horizontal fibre direction. The micrographs of the composites with vertical and random fibre orientations exhibited ductile fractures, while that with horizontal and cross fibre orientations showed brittle fractures. Considering its nice fragrance, Masdani, et al. [17] studied the development of agarwood fibre-reinforced composites as a substitute for fibreglass in the manufacture of car dashboards using the hand lay-up technique. The highest tensile strength (34.574 MPa) was found at fibre volume fraction (Vf) of 45%. Scanning Electron Microscope (SEM) images of the fracture section of specimens with a volume ratio of 40% and 50% showed fibre pull-out indicating imperfect fibre-matrix interfacial bonding. Witayakran, et al. [18] reported that although the incorporation of oil palm empty fruit bunch fibre into epoxy can prevent the bumper beam specimen from breaking apart as in that of neat epoxy, its flexural modulus was not affected by the fibre content up of 7.5 wt%. A previous study compared the characteristics of CFRP composite between those produced using manual layup and vacuum infusion methods at a fibre weight fraction (Wf) of 60% [19]. It was found that the strength of the CFRP composite produced using the vacuum infusion method is higher than that produced using the manual lay-up method. The vacuum infusion method resulted in the average tensile strength and modulus of 595.63 MPa and 6.976 MPa, respectively; as well as the average impact strength and absorbed energy of 33.25 J/cm$^2$ and 2,433.65 J, respectively. The lay-up method resulted in the tensile strength and modulus of 581.93 MPa and 10.241 GPa, respectively, as well as the average impact strength and absorbed energy of 29.41 J/cm$^2$ and 2212.59 J, respectively. More evenly distributed fibre leading to more homogeneous properties of the resulted composites produced using vacuum infusion may be responsible for such result.

The durability of composite materials in an aqueous environment is largely determined by the amount of water being absorbed from the environment either in the form of liquid or vapour. To determine the effect of CFRP exposed to a humid environment causing water intake, several studies have been conducted and reported. Autay et al. [20] reported that although flexural strength and modulus of short glass fibre-reinforced nylon 66 composites decrease due to hygrothermal aging, its wear resistance and coefficient of friction increase. The failure mode of the flexural specimens was not discussed. Almudaihesh et al. [21] reported that the mechanical properties of CFRP composite decreased due to water absorption. The magnitude of the reduction depends on the fibre architecture and the loading case, where unidirectional fibre arrangement resulted in the largest reduction. Although failure modes have been discussed, more detail about the failure mechanism cannot be assessed due to the low magnification of the images being analyzed. The present study investigates the effect of water absorption on CFRP laminate mechanical properties, i.e., tensile, compressive, and flexural properties. While a previous report [21] used low magnification images to assess failure mode, the current study used higher magnification images captured using an optical microscope resulting in more detailed information regard-

ing the failure can be analyzed. This study can be used to determine the behaviour and the physical effect of the CFRP that in the future can be applied in different applications.

## 2. Materials and Methods

### 2.1. Materials

The matrix being used in this work is Eposchon A epichlorohydrin epoxy resin combined with Eposchon B polyaminoamide epoxy hardener, and the reinforcement is 200-3K 2/2 twill weave carbon fibre. Both were purchased from a local supplier, PT Justus Kimia Raya. The reinforcement was made from polyacrylonitrile precursor and was produced and imported from China. Important physical and mechanical properties of the matrix and the reinforcement collected from their respective data sheets and related references have been presented in Table 1. It should be noted that epoxy, the matrix used in this current work, does not follow the Fickian pattern in relation to moisture absorption [22].

**Table 1.** Mechanical properties of constituent materials.

| Properties | Matrix [23] | Fibre [24] |
|---|---|---|
| Viscosity (cP) | 13,000 a, 12,000 b | data |
| Density at 25 °C (g/cm3) | 1.15 a, 0.97 b | 1.78 ± 0.06 [25] |
| Weave | - | Twill |
| Width (mm) | - | 1500 ± 5 |
| Area weight (gr/m2) | - | 200 ± 5 |
| Thickness (mm) | - | 0.28 ± 0.02 |
| Weave Density, warp (ends/100 mm) | - | 50 ± 2 |
| Weave Density, wave (ends/100 mm) | - | 50 ± 2 |
| Breaking strength, warp (N/25 mm) | - | >1000 |
| Breaking strength, weft (N/25 mm) | - | >1000 |
| Tensile strength (MPa) | ~48.68 [26] | 4050 ± 150 |
| Compressive strength (MPa) | ~89.15 [27] | - |
| Tensile modulus (GPa) | ~1.75 | 211 ± 9 |

[a] resin, [b] hardener.

### 2.2. Composite Plate Fabrication

Specimens for physical and mechanical characterisations were cut from composite plate panels fabricated using vacuum infusion technique whose lay-up is shown in Figure 1. Fibre content was kept constant at ~40 vol%.

A schematic illustration of the mould detail has been presented in Figure 1b. To ease of taking off the plate from the base plate, the surface of the base plate was coated using silicone mould release by spraying it onto the surface. After it was dried, mirror glaze was applied on the above surface of the coating using a very soft cloth such that a smooth surface of the plate could be obtained. Following these, several layers of twill weave carbon fibre as required, a layer of peel ply and flow media were carefully arranged on it. A hose T-connected to a spiral hose was placed at one edge of the carbon fibre arrangement for resin inlet and distribution. A similar hose T-connected to a spiral hose was also placed at the opposite edge of the former for sucking out the air from the mould cavity, sucking in resin from the resin container for wetting the fibre arrangement, and sucking out excess resin. All of these arrangements were then covered with bagging film, and air-tightly sealed from the outer environment using sealing tape. It should be noted that both the inlet and outlet hoses have to be airtight as well.

Prior to being used, the set-up in Figure 1a must be guaranteed for no leakage. This can be tested by tightening the inlet clamp (9b), sucking out the air inside the mould cavity (5) until reaching the required vacuum level, tightening the outlet clamp (9a), switching off the vacuum pump and waiting for approximately 10 min. After ~10 min, check the vacuum gauge. If the vacuum level remains constant then the seal is perfect, and prepare the required resin-hardener mixture. Otherwise, find the leakage and fix it before mixing the resin with the hardener. Resin-hardener mixing ratio as recommended by the manufacturer

is 1:1 by weight. When the mixture was ready, switch on the vacuum pump and open the inlet clamp, then the resin will be drawn into the mould cavity due to vacuum pressure inside the cavity. When the fibre arrangement was completely wet, excess resin containing a lot of air bubbles can be seen flowing through the outlet hose and entering the excess resin container (7). Curing took between 12 to 16 h before the plate could be taken off the base plate surface.

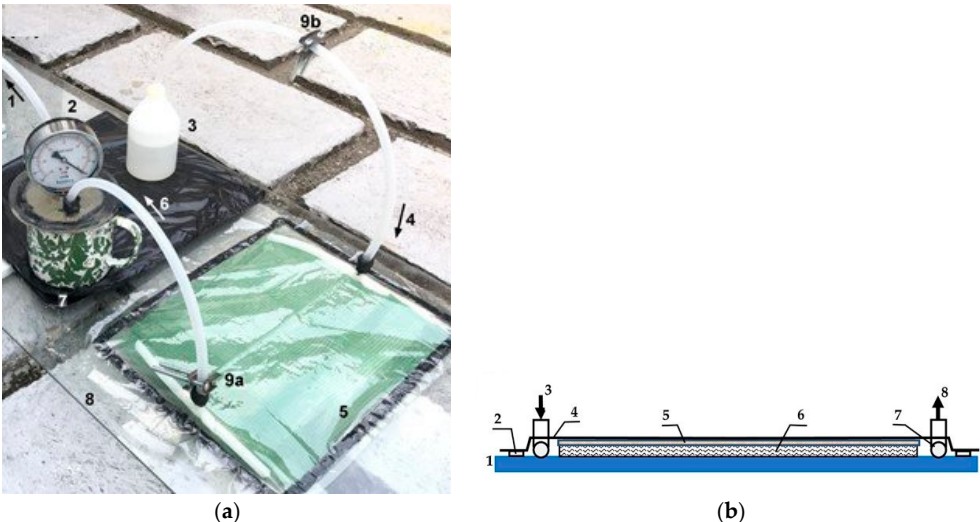

(a)                                   (b)

**Figure 1.** Composite plate fabrication by vacuum infusion technique: (**a**) Set up: 1. To vacuum pump, 2. Vacuum gage, 3. Resin container, 4. Resin inlet, 5. Mould arrangement, 6. Excess resin outlet, 7. Excess resin container, 8. Base, thick glass plate, 9a. Outlet clamp, 9b. Inlet clamp; (**b**) Schematic illustration of mould arrangement detail: 1. Glass plate base, 2. Air-tight sealer, 3. Resin inlet, 4. Bagging film, 5. Peel ply and flow media, 6. Fibre arrangement, 7. Outlet tube, 8. Excess resin outlet.

## 2.3. Specimen Preparations

Specimens were cut from the composite plates produced by means of the vacuum infusion molding technique as has been outlined above. They were cut using a diamond-tipped circular blade saw rotating at ~6000 rpm. While tensile, compressive and flexural test specimens were prepared according to the ASTM D638 [28], ASTM D695 [29] and ASTM D790 [30], respectively, those of water absorption test were prepared according to the ASTM D570 [31]. To avoid edge effects due to cutting, the specimens were fine-polished before being tested. The geometries of the specimens according to their respective adopted standards have been presented in Figure 2.

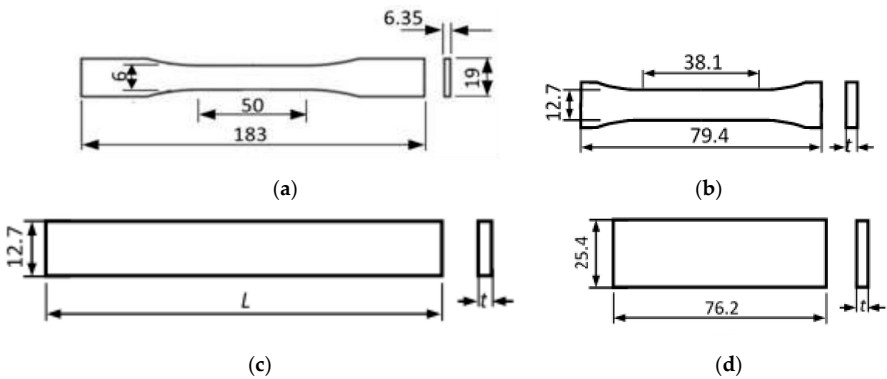

(a)                                   (b)

(c)                                   (d)

**Figure 2.** Specimen geometries according to the adopted standards: (**a**) tension (Type II), (**b**) compression ($t = 3.2 \pm 0.1$ mm), (**c**) flexure ($L = 77$ mm, $t = 3.2 \pm 0.1$ mm), and (**d**) water absorption ($t = 3.2 \pm 0.1$ mm). **Notes**: $t$ depends on the thickness of the plate from which they were cut.

### 2.4. Water Absorption Test

Unlike that previously reported [21] where immersion was conducted at 70 °C for 40 days based on ASTM D5229, considering most vehicle parts are exposed to rain for quite a short of time, immersion in the current work was carried out according to the ASTM D570 at room temperature for $8 \times 6$ h. The data presented here are the average values of the six specimens being tested. The specimens were immersed in fresh water for 48 h at room temperature, where they were taken out, wiped using a dry cloth and weighed every 6 h using a digital scale possessing an accuracy of 0.010 g. The weight gains of the specimens were recorded, and their respective percentage was calculated using the equation ad the thickness increase using (1) below.

$$W_{\mathrm{A}} = \frac{m_1 - m_2}{m_1} \cdot 100\% \tag{1}$$

where $W_{\mathrm{A}}$, $m_1$ and $m_2$ are water absorption (%), initial mass [g], and final mass [g], respectively.

### 2.5. Tensile Test

The ASTM D638 tensile test standard was adopted as a reference, and the specimens were of Type II. They were tested at room temperature using a Gotech GT-7001-LC50 Universal Testing Machine (UTM), as can be seen in Figure 3a. The specimens were tested at constant strain rate of 1% per minute. The number of specimens being tested is six for each case as recommended by the standard, and the data presented here are their average values. Tensile strength and strain were calculated as per their respective definitions. It should be noted here that extension of the specimens was measured based on the displacement of the UTM crosshead travels after being corrected for the nonlinear parts at the initial stages. It may be caused by the tightening effect of the gripping.

### 2.6. Compressive Test

Compressive test was carried out in accordance with the ASTM D695. They were tested at room temperature using a Zwick Roell Z020 Universal Testing Machine (UTM), as presented in Figure 3c, in the ATMI Polytechnique, Surakarta. The specimens were tested at a constant straining rate of 1.3 mm/min

The number of specimens being tested is six for each case as recommended by the standard, they were then averaged and presented here. While compressive modulus was calculated using Equation (2) at a strain range of 0.3–0.5% where the magnitudes of stress and strain were taken from the initial linear portion of their respective stress-strain curves [29], compressive strength and strain were calculated according to their respective definitions. The contraction of the specimens was measured based on the displacement of the UTM crosshead travels after being corrected for the nonlinear parts at the initial stages. It may be caused by unavoidable specimen negligible misalignment of the specimens.

$$E_{\mathrm{c}} = \Delta\sigma / \Delta\varepsilon \tag{2}$$

where $E_{\mathrm{c}}$ is compressive modulus [MPa].

### 2.7. Flexural Test

The flexural test was conducted in accordance with the ASTM D790 standard in which the specimens were tested at room temperature using a Zwick Roell Z020 UTM, as depicted in Figure 3b, in the ATMI Polytechnique, Surakarta. The specimens were tested at a constant strain rate of ~1% per minute. The number of specimens being tested is six for each case as recommended by the standard, and the data presented here are their average values. There were two cases in this work, i.e., dry specimens and wet or after-immersed specimens. To ensure that the specimens fail by flexure instead of by shear, the adopted standard recommends that a span-to-depth ratio, S/d, of 16 or larger should be used. Considering

this, the S/d being used in this research is ~24. Flexural stress, strain and modulus were calculated using (3a), (3b) and (3c), respectively [30].

$$\sigma_\mathrm{f} = \frac{3FS}{2bd^2} \cdot \left( 1 + 6\frac{D^2}{S^2} - 4\frac{dD}{S^2} \right) \tag{3a}$$

$$\varepsilon_\mathrm{f} = \frac{6dD}{S^2} \tag{3b}$$

$$E_\mathrm{f} = \frac{S^3 m}{4bd^3} \; ; \quad m = \Delta F / \Delta D \tag{3c}$$

where

$\sigma_\mathrm{f}$ = flexural stress at the point being observed (MPa)
$F$ = the magnitude of load (N)
$S$ = support span (mm)
$b$ = width of the beam (mm)
$d$ = depth of the beam (mm)
$D$ = mid-point deflection where the load being applied (mm)
$\varepsilon_\mathrm{f}$ = flexural strain at the point being observed (mm/mm)
$E_\mathrm{f}$ = flexural modulus (MPa)
$m$ = slope of the initial linear portion of F-D curve (N/mm)
$\Delta$ = indicates the magnitude of the quantity within the range being observed.

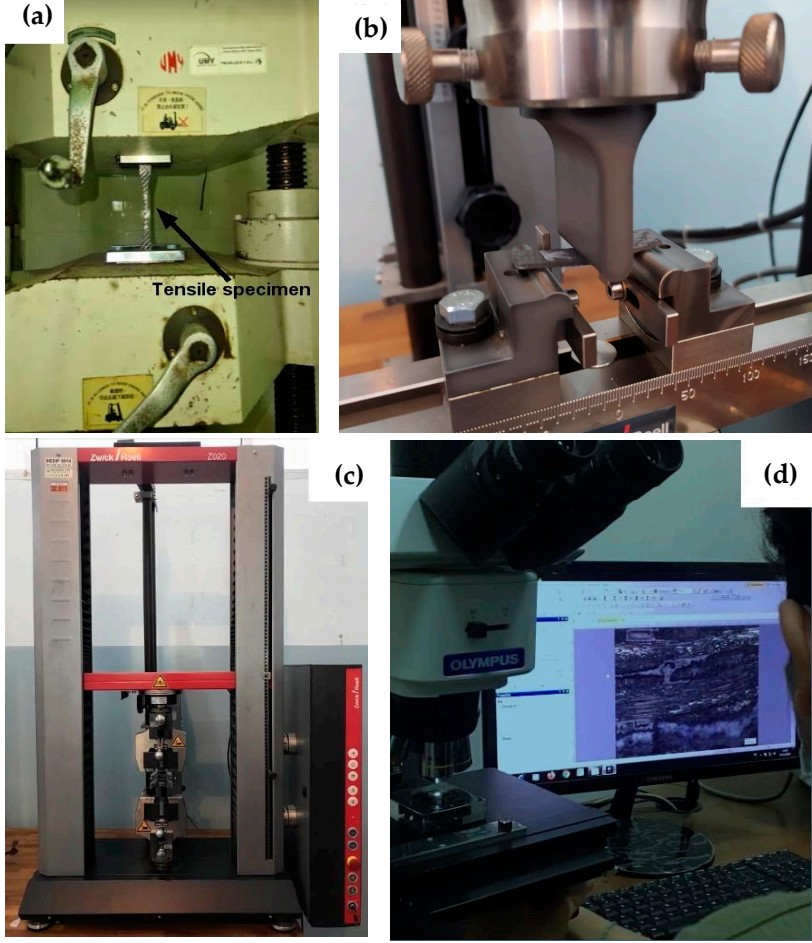

**Figure 3.** Experimental devices: (**a**) gripping device for tensile test, (**b**) Zwick Roell Z020 used for compression and flexural test, (**c**) flexural test fixture, (**d**) Olympus BX53M used for image capturing.

*2.8. Failure Mode Evaluation*

Although work on compressive failure mode has previously been reported [21], it was based on analysis of low magnification images resulting in a rough failure mechanism description. Failure mode analysis in the current work was based on much higher magnification images resulting in more detailed failure mechanism. After being mechanically tested, some broken specimens of each case were randomly selected for image-capturing preparation. The broken regions of the selected specimens were cut into approximately 20 mm long, casted in polyester resin blocks, and mirror polished. These samples were then, one by one, put under an Olympus BX53M Microscope lens, Figure 3d, for image capturing. Failure modes were evaluated by closely observing and analysing their respective micrographs.

## 3. Results

### 3.1. Water Absorption

Water absorptions were calculated using (1). The data obtained from the immersion of the specimens for 48 h has been presented in Figure 4. The figure shows that the saturation has not been reached after the immersion. Sugiman et al. [26] found that epoxy matrix composites containing different fillers of Portland cement, fly ash, and $CaCO_3$ exhibited water uptake saturation after being immersed in distilled water at 50 °C for approximately three days. In addition, an increase in the content of reactive filler resulted in an increase in water uptake, and vice versa for nonreactive fillers. Thus, their water uptake also varies between ~3.5% for $CaCO_3$ particle-reinforced epoxy composites at fibre volume fraction, $V_f$, of 25% to above 18% for Port-land cement-reinforced epoxy composites at $V_f$ = 25%.

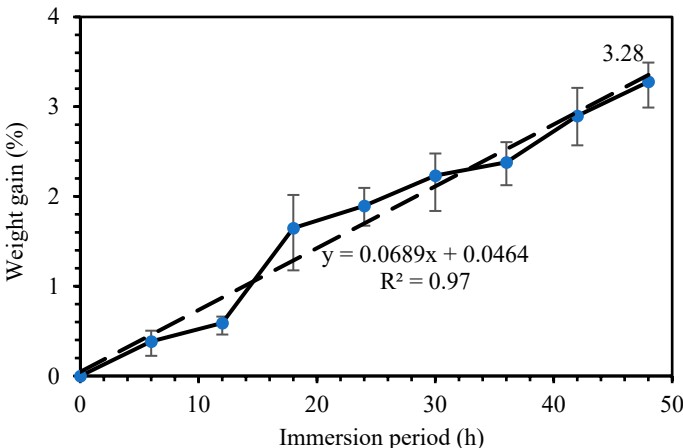

**Figure 4.** Water absorption pattern.

### 3.2. Tensile Properties

#### 3.2.1. Stress-Strain Relation

Apart from specimen number 2 in Figure 5a, Figure 5 shows that tensile stress-strain relations exhibited considerably linear. This is consistent with the theory where mechanical properties in fibre direction are strongly determined by fibre properties that possess linear stress-strain relation up to fracture [32]. In addition, their peak loads dan slopes are highly similar from one to the other. Thus, it was expected that their tensile strength, strain-to-failure, and elastic modulus were also varies in a narrow spread with short error bars, even between dry specimens on one hand and wet specimens on the other hand. The increase in the slope was suspected caused by straightening of wavy fibres.

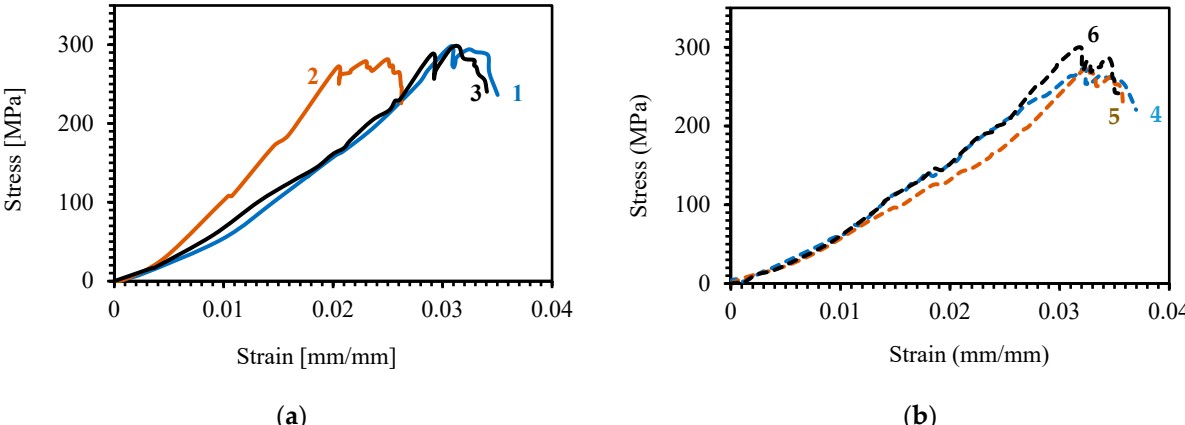

**Figure 5.** Tensile stress-strain relation: (**a**) dry specimens 1, 2 and 3; (**b**) wet specimens 4, 5 and 6.

### 3.2.2. Tensile Strength

The effect of water swelling on tensile strength has been presented in Figure 6. It can be observed that a slight decrease in tensile strength, approximately 3.52% with respect to that of dry specimens, due to water absorption can be observed in Figure 6a. According to Shetty et al. [33], the degradation of mechanical properties of their GFRP samples is linearly proportional with the degree of moisture content. This is caused by the decrease of fibre-matrix interfacial strength as has been reported by Eslami et al. for GFRP composites [34].

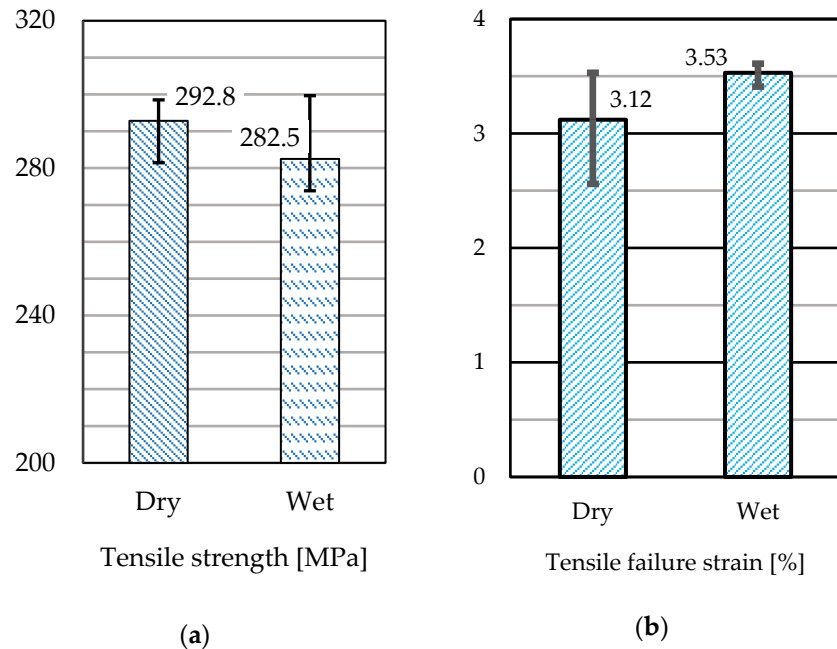

**Figure 6.** Tensile properties: (**a**) Tensile strength, (**b**) Tensile failure strain.

### 3.2.3. Tensile Failure Strain

In contrast to tensile strength, the tensile failure strain was found being slightly increase due to water swelling. A weaker fibre-matrix interfacial bonding will result in easier matrix to deform leading to higher strain-to-failure. This contrasts with GFRP composite as reported by Idrisi et al. [35].

### 3.2.4. Tensile Failure Mode

Figure 7 shows photo macrographs of fracture tensile specimens. Figure 7a represents dry specimens, dan Figure 7b represents wet specimens. Light grey dots in both figures are

cross-sections of weft fibres. It can be seen that out-of-plane weaviness due to the weaving of warp fibres almost cannot be seen.

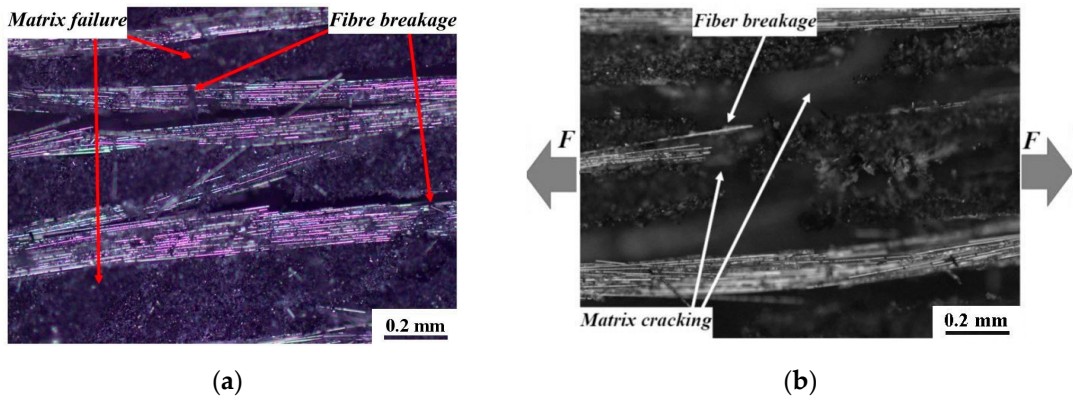

(**a**)          (**b**)

**Figure 7.** Tensile failure modes: (**a**) dry specimens, (**b**) wet specimens.

*3.3. Compressive Properties*

3.3.1. Compressive Stress-Strain Relation

Figure 8 shows that tensile stress-strain relations are noticeably linear up to failure. Following the first load drop, the plot shows a relatively constant magnitude of the load. This may be caused by the compaction of the matrix after the fibre could no longer be able to take any parts of the load. The three samples demonstrated homogeneous peak loads and slopes suggesting that their compressive strength, strain-to-failure, and elastic modulus would also vary in a narrow spread, even between dry specimens on one hand and wet specimens on the other hand. This initial nonlinear part was termed as "toe" in the ASTM D790 standard [30], and it should be corrected or compensated.

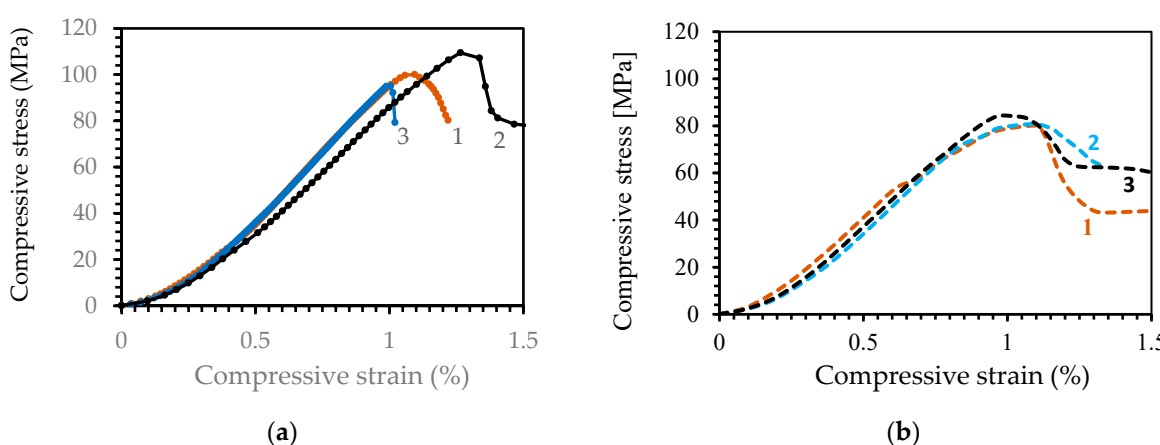

(**a**)          (**b**)

**Figure 8.** Compressive stress-strain relation: (**a**) dry specimens 1, 2 and 3; (**b**) wet specimens 1, 2 and 3.

3.3.2. Compressive Strength

The effect of water uptake on compressive properties has been presented in Figure 8. It was found that compressive strength (102.3 MPa) was only ~35% of the tensile strength (292.8 MPa). This may be caused by its fibre kinking followed by crushing failure mode (Figure 10) which is initiated by matrix shear deformation [36]. Figure 9a shows a significant decrease in compressive strength, approximately 21.02% with respect to that of dry specimens, can be observed. As has been discussed earlier water uptake can degrade fibre-matrix interfacial bond strength causing a reduction in mechanical properties.

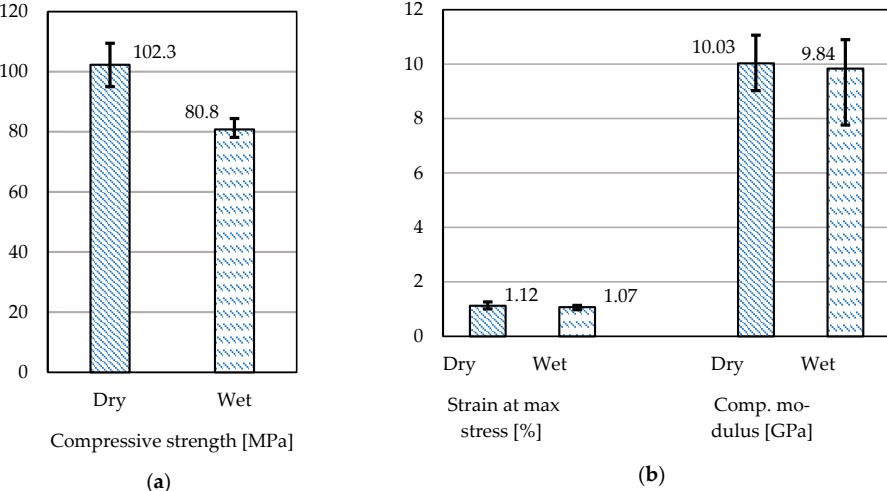

**Figure 9.** Compressive properties: (**a**) Compressive strength, (**b**) Strain at initial failure and modulus.

### 3.3.3. Compressive Fracture Strain

As can be seen in Figure 9b, fracture strain was found being 1.12% and 1.07% for CFRP dry and wet specimens, respectively, which is significantly higher than that previously reported for dry CFRP specimens, ~0.62% [37]. Unlike compressive strength undergoing significant decrease, Figure 9b shows that the decrease of fracture strain is much lower, only ~4.46%. In contrast to compressive strength, the fracture strain was found to only a slight decrease due to water swelling.

### 3.3.4. Compressive Modulus

Compressive modulus was calculated using (2) of the initial linear part of plots in Figure 8 as recommended by the adopted standard, and presented in Figure 9b. These moduli, 10.03 GPa for dry specimens and 9.84 GPa for wet specimens, are significantly lower than those previously reported for unidirectional CFRP composites, ~104.5 GPa for dry specimens [37].

### 3.3.5. Compressive Failure Modes

Figure 10 shows macrographs of fractured compressive specimens. It can be seen that fracture was initiated by fiber buckling followed by kinking due to matrix shear deformation [38] and out-of-plane fibre waviness. Further compression would lead to fibre fracture and matrix crushing. There is no significant different in compressive failure mode between dry specimens and wet specimens can be noticed.

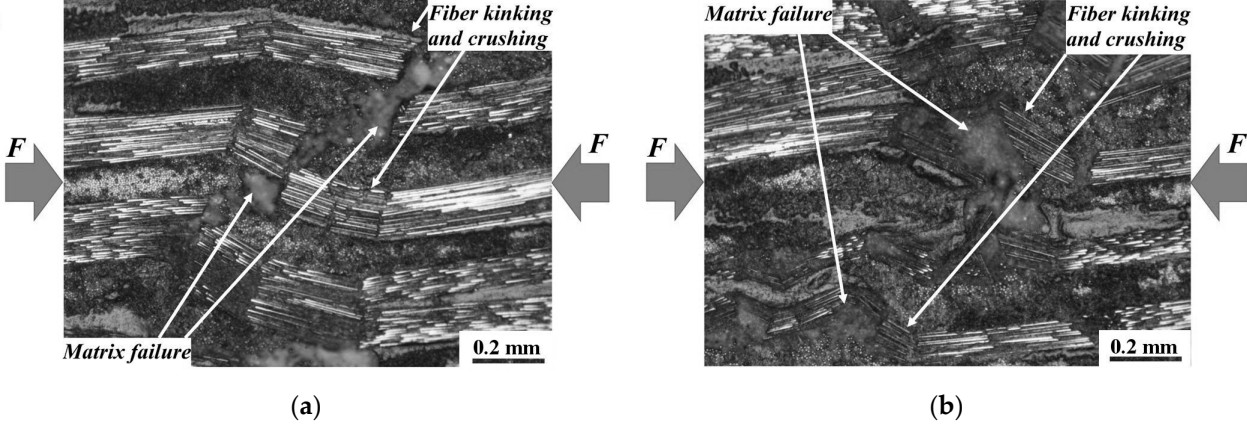

**Figure 10.** Compressive failure modes: (**a**) dry specimens, (**b**) wet specimens.

*3.4. Flexural Properties*

3.4.1. Flexural Stress-Strain Relation

Flexural stress and strain were calculated using (3a) and (3b), respectively. Flexural stress-strain relations of the samples have been presented in Figure 11. Those of the dry specimens, as shown in Figure 11a, show a relatively small variation in slopes but with a considerably large peaks. While the peaks were found at 0.01 mm/mm strain, plastic deformation continued even up to more than 0.06 mm/mm strain. Matrix compaction due to compression after fracture may be responsible for this deformation. Unlike those of dry specimens, those of wet specimens in Figure 11b look more similar to the other in terms of their slopes and peak loads. Comparing Figures 10b and 11a suggests that there will be a small decrease in magnitudes of flexural modulus and strength due to moisture content similar to the tensile modulus and strength of CFRP composites reported by Asasaari et al. [36]. It should be noted that all the specimens were failed initiated at the compression side propagated through the thickness, similar to those previously reported [39].

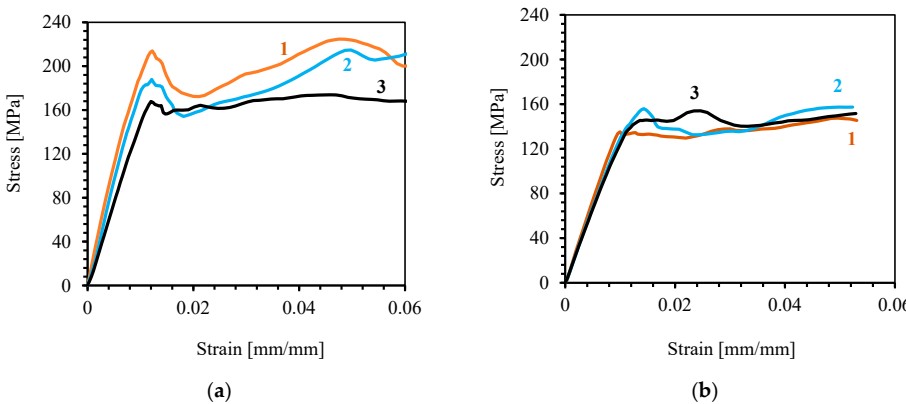

**Figure 11.** Flexural stress-strain relation: (**a**) dry specimens 1, 2 and 3; (**b**) wet specimens 1, 2 and 3.

3.4.2. Flexural Strength

Figure 12 shows the effect of moisture content on the flexural properties of the samples. Flexural strength of dry specimens (196.2 MPa) was found being 39.76% higher than that of GFRP composites (~127.5 MPa) as reported by Mukhtar et al. [40] owing to the strength of carbon fibre (4050 MPa, Table 1) being significantly higher than that of E-glass fibre (2400 MPa) [41]. Figure 12(a) shows a significant (25.74%) decrease in flexural strength at 3.28% moisture content.

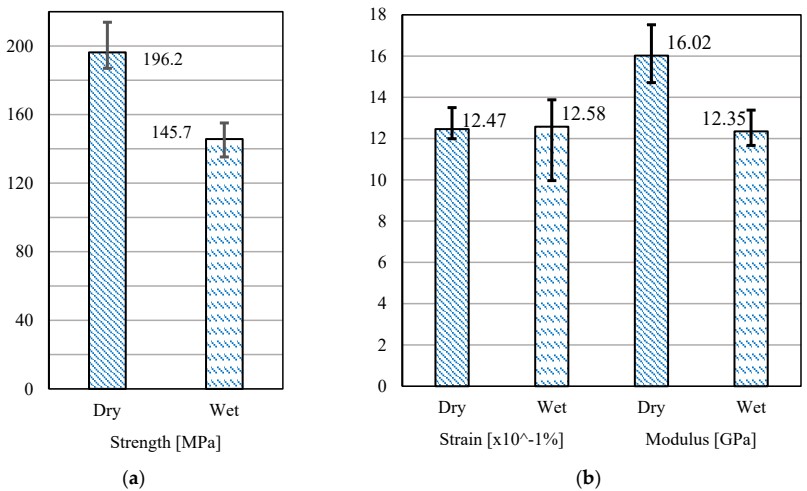

**Figure 12.** Flexural properties: (**a**) Flexural strength, (**b**) Flexural fracture strain and modulus.

### 3.4.3. Flexural Fracture Strain

It can be seen in Figure 12b that similar to tensile strain, the flexural fracture strain of wet specimens was found being slightly higher than that of dry specimens due to water swelling. Again, a weaker fiber-matrix interfacial bonding will allow the matrix to deform more freely leading to larger elongation and higher fracture strain. Furthermore, a significant decrease in flexural strength combined with only a slight increase in flexural fracture strain will tend to result in a decrease in flexural modulus.

### 3.4.4. Flexural Modulus

Flexural modulus calculated using (3c) was presented in Figure 6b. These moduli, 16.02 GPa and 12.05 GPa for dry and wet specimens, respectively. The flexural modulus of dry specimens is ~24.19% higher than those previously reported for woven E-glass fibre-reinforced polypropylene composites (12.9 GPa) [40]. In addition, Figure 12b also shows that the flexural modulus of wet specimens decreased by approximately 22.91% with respect to that of dry specimens.

### 3.4.5. Flexural Failure Modes

Figure 13 shows macrographs of the compressive side of fracture flexural specimens. Figure 13a,b shows fibre kinking and crushing. Out-of-plain fibre warp fibre waviness, micromechanically causing excentric loading causing premature fibre buckling as happened in compressive specimens [38] resulting in low flexural strength. Wet specimens, Figure 13b, show more severe damage compared to Figure 13a does. This may be cause why the magnitude of strength and modulus of wet specimens are lower than those of dry specimens.

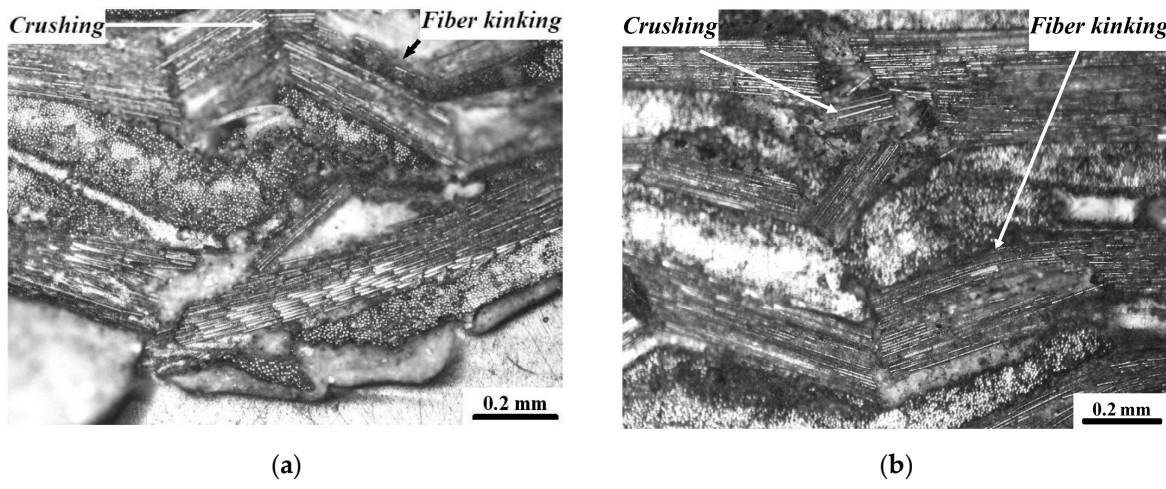

(**a**)　　　　　　　　　　　　　　　　　　(**b**)

**Figure 13.** Flexural failure modes: (**a**) dry specimens, (**b**) wet specimens.

## 4. Discussion

Idrisi et al. [32] found that E-glass/epoxy composites being exposed in seawater at various temperatures exhibited saturation at approximately 25–30 months, where their water uptake at saturation was higher at a higher temperature of immersion, i.e., 7% at 65 °C. This lower water uptake at saturation and a lower rate of water absorption supports what Sugiman et al. [26] reported. The current result is also in a good agreement with previous findings [26,32] where the water uptake is reasonably low combined with the positive trend of water absorption at a longer immersion period.

Pai et al. [41] reported that the tensile strength and modulus of basalt-aramid/epoxy hybrid composites decreased by 21.88% and 18.83%, respectively, due to water aging at room temperature. The decrease in tensile strength was found being significantly lower than those of compressive and flexural strength. It is commonly known that tensile strength is fiber-dominated property, whilst compressive strength is more matrix-dominated. In

the case of flexure where specimens fail on the compressive side also shows a significant decrease in flexural strength. Similarly, a previous report by Li [42] showed a significant decrease in compressive strength. Straightening of wavy fibres due to waving prior to being fully loaded up to fracture resulting in higher strain-to-failure also contributed to a lower modulus of woven CFRP composites compared to that of unidirectional CFRP composites previously reported [36]. No significant difference in tensile failure mode between dry specimens and wet specimens was observed.

The lower compressive modulus of woven CFRP composite compared to that of unidirectional CFRP composites may be attributed to the out-of-plane fibre waviness causing microscale compressive loading instability resulting in premature buckling and fibre kinking, as can be seen in Figure 10, lower strength [43] and larger failure strain leading to lower modulus. In addition, Figure 9b also shows a slight decrease in compressive modulus due to moisture content. In the case of flexural properties, considering flexural failure was initiated at the compressive side as shown in Figure 13, it can be inferred that the flexural properties of these CFRP samples also strongly depends upon their respective compressive properties similar to those previously reported for CFRP composites [44,45]. The decrease in flexural strength (25.74%) is slightly higher than that of compressive strength (21.02%). In other words, flexural strength is severely affected by swelling. The compressive modulus of wet twill woven CF-reinforced epoxy composite is sensitive to moisture due to its low bending stiffness owing to its small fibre diameter [38] and matrix degradation due to water uptake. Considering parameters of composite materials affect their properties related to water absorption, further research on the effect of different fiber architectures, reinforcement-matrix combinations, and fiber geometries on water absorption pattern and their performance need to be investigated.

## 5. Conclusions

Water content badly affects the strength of the composites, where compressive and flexural strength are found being more pronounced in comparison with that of tension, decreased by 21.02% and 25.74%, respectively, compared to a 3.52% decrease in tensile strength. Unlike the strength of the composites, fracture strain was found to increase due to water swelling. Because of the decrease in strength and increase of fracture strain, modulus was found being significantly decreased due to water uptake. The decrease in flexural modulus (22.91%) was found being much higher than that of compressive modulus (1.89%). Tensile specimens underwent fibre breakage followed by matrix cracking, compressive and flexural specimens showed fibre buckling followed by kinking and crushing.

**Author Contributions:** Conceptualization, S.S.; methodology, S.S. and R.A.H.; validation, M.A.M.; formal analysis, S.S. and R.A.H.; investigation, H.H., A.K.T. AND M.T.; resources, H.H., A.K.T. AND M.T.; data curation, S.S. and M.A.M.; writing—original draft preparation, S.S.; writing—review and editing, M.A.M.; visualization, R.A.H.; supervision, S.S.; project administration, R.A.H. All authors have read and agreed to the published version of the manuscript.

**Funding:** This research received no external funding.

**Data Availability Statement:** Data analyzed in this study are available and can be found at https://umyac-my.sharepoint.com/personal/sudarisman_umy_ac_id/_layouts/15/onedrive.aspx?id=%2Fpersonal%2Fsudarisman%5Fumy%5Fac%5Fid%2FDocuments%2F2023%2E02%2E25%20water%20absorption%20data&ga=1 (accessed on 8 March 2023).

**Acknowledgments:** The authors would like to thank the Industrial Machinery Technology Academy (ATMI), Surakarta for testing the compressive and flexural specimens.

**Conflicts of Interest:** The authors declare no conflict of interest.

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
