# Peer review of "Tensile, Compressive, and Flexural Characterization of CFRP Laminates Related to Water Absorption"

_jcs, doi:10.3390/jcs7050184_

Round 1

Reviewer 1 Report

This paper studies water swelling and its effect on tensile, compressive, and flexural properties, which has potential application value in engineering. In order to meet the requirements of high-quality publication of the journal, it is recommended to consider the following suggestions,

1) There is no quantitative data in Abstract Section.

2) The innovation of this article is not reflected in the first section and needs to be modified.

3) What is "Figure (a): 1. To vacuum pump, 2. Vacuum gage," in Line 114?

4) The second Section needs to add pictures of the experimental device.

5) The mehtod proposed in this paper needs to be compared with the previous literature, otherwise it cannot reflect innovation.

6) The Discussion Section needs a separate section.

7) There is no quantitative data in the Conclusion Section.

8) There are few references in the last three years.

Author Response

Thank you for your the manuscript improvement. Discussion has been separated from Result, thus the lay uot, especially the last number of pages, has changed quite significantly.

Please find the responses to your comments as appended.

Reviewer 2 Report

This is another experimental article about FRP (namely, carbon FRP). The possibilities of experimental technology grow, as a result, there are more and more high-quality (in the sense of conducting an experiment, not in the sense of ideas) experiments. Should such work be published? Maybe yes. Qualitative experiment can lead someone to new ideas, or provide material for thinking about or testing new ideas. Whether this article will stimulate someone to new ideas, we will find out in 10 years.

Now we can say that the article contains experimental data, the data are qualitative. And I hope that the article will be useful to someone.

Author Response

Thank you for your invaliable comment for the manuscript improvement. Please note that the lay out has changed quite significantly as suggested by one the three Reviewers.

Plesae find the appended responses to your comments.

Reviewer 3 Report

The paper provides useful results on mechanical properties of CFRP 2 Laminates. However, it should be improved for the following aspects:

·        Figure 2 is a mere reproduction of standard drawings. The actual range of values of L and t for the respective testing should be provided here.

·        Section 5.2: the authors should remind whether an extensometer was applied to measure the strain, and provide details of the model used

·        Section 5.3: as above

·        Section 3.1.1: the authors claim a linear relationship, however, following the range where the elastic modulus is calculated, the slope of the curves increases. The authors should comment about this trend, is it expected?

·        Section 3.3.1: the initial range is highly non-linear. The authors should comment whether this is due to specimen sitting or other causes. Moreoevr the authors say “compressive modulus was calculated using Equation (3) at strain range of 0.5-1.5 %” but 1.5 % is already beyond rupture based on Fig. 7!

Author Response

(The authors gave the same response as above.)

Round 2

Reviewer 1 Report

The authors have addressed all my concerns.

Author Response

Thank you very much for accepting the revision and approving for further process.

Reviewer 3 Report

It is highly questionable the measurement of tensile elastic moduli without the use of an extensometer. In particular, standard ASTM D638 imposes that for modulus of elasticity measurements, an extensometer with a maximum strain error of 0.0002 mm/mm (in./in.) that automatically and continuously records shall be used and reminds that an extensometer classified by Practice E83 as fulfilling the requirements of a B-2 classification within the range of use for modulus measurements meets this requirement. Now, the authors have responded that the extension was measured based on the travel of the crosshead, which is unacceptable. New tests should be carried out according to ASTM D638 that the authors claim as a reference for the above tests.

Author Response

The authors highly appreciated your invaluable comment regarding accessing tensile modulus.

Round 3

Reviewer 3 Report

The authors have taken into account the major remarks formulated and in my view the present version can be accepted pending some English text editing.